# Domain-specific Benchmarking of Vision-Language Models: A Task Augmentation Framework Using Metadata

## Abstract

The reliable and objective evaluation of AI models is essential for measuring scientific progress and translating methods into practice. However, in the nascent field of multimodal foundation models, validation has proven to be even more complex and error-prone compared to the field of narrow, task-specific AI. One open question that has not received much attention is how to set up strong vision language model (VLM) benchmarks while sparing human annotation costs. This holds specifically for domain-specific foundation models designed to serve a predefined specific purpose (e.g. pathology, autonomous driving) for which performance on test data should translate into real-life success. Given this gap in the literature, our contribution is three-fold: (1) In analogy to the concept of data augmentation in traditional ML, we propose the concept of task augmentation - a resource-efficient method for creating multiple tasks from a single existing task using metadata annotations. To this end, we use three sources to enhance existing datasets with relevant metadata: human annotators (e.g. for annotating truncation), predefined rules (e.g. for converting instance segmentations to the number of objects), and existing models (e.g. depth models to compute which object is closer to the camera). (2) We apply our task augmentation concept to several domains represented by the well-known data sets COCO (e.g. kitchen, wildlife domain) and KITTI (autonomous driving domain) datasets to generate domain-specific VLM benchmarks with highly reliable reference data. As a unique feature compared to existing benchmarks, we quantify the ambiguity of the human answer for each task for each image by acquiring human answers from a total of six raters, contributing a total of 162,946 human baseline answers to the 37,171 tasks generated on 1,704 images. (3) Finally, we use our framework to benchmark a total of 21 open and frontier closed models. Our large-scale analysis suggests that (I) model performance varies across domains, (II) open models have narrowed the gap to closed models significantly, (III) the recently released Qwen2 72B is the strongest open model, (IV) human raters outperform all VLMs by a large margin, and (V) many open models (56%) perform worse than the random baseline. By analyzing performance variability and relations across domains and tasks, we further show that task augmentation is a viable strategy for transforming single tasks into many and could serve as a blueprint for addressing dataset sparsity in various domains.

## 1 Introduction

The reliable and objective performance assessment, i.e., validation of AI models is crucial for both the measurement of scientific progress and translation into practice. Benchmarking for traditional narrow, task-specific AI already comes with numerous challenges (Myllyaho et al., 2021), but validation has proven to be even more complex and error-prone in the emerging field of generalist multimodal foundation models. For example, an award-winning Neurips paper (Schaeffer et al., 2024) on language foundation models recently showed a large discrepancy between progress claimed by researchers and fundamental changes in model behavior with scale. In the context of vision language models (VLMs), benchmarking-related issues include data leakage (Chen et al., 2024a), a

narrow task focus with inadequate coverage of vision-language capabilities, labor-intensive and time-consuming human annotation, and frequent failure to report the number and qualifications of human annotators, as well as potential disagreements in their annotations. One open question that has not received much attention is how to set up strong domain-specific VLM benchmarks while sparing human annotation costs. This paper therefore focuses on the specific task of VLM validation from the perspective of (annotation) resource investment. It is based on the following key observations:

1. *Cross-domain validation is not always desirable.* While numerous datasets and benchmarks are currently being released in the general computer vision field (e.g. 400 out of the 2700 CVPR 2024 publications propose a new or modified dataset, see Appendix C), these may not always be optimal for validating domain-specific foundation models designed to serve a predefined specific purpose. In such cases, performance on test data should translate into real-life success, thus requiring the test set to represent the challenges and variability found in the specific domain the model was designed for (e.g., medicine). Given this, arena-style evaluations—where users submit tasks and rate models blindly—provide strong evidence for the need for domain-specific evaluation by ensuring task-relevant assessments, such as Chatbot Arena[1] or WildVision Arena[2]. Domain-specific evaluation, however, comes with challenges, as publicly available data may be sparse and human resources for labeling can be limited (see e.g. Maier-Hein et al. (2022)). An open question, therefore, is how to make the best out of available data and resources.

2. *Picking the right tasks is challenging.* Foundation models should be adaptable to a wide variety of domain tasks. The challenge in traditional computer vision was to obtain a highly diverse set of representative images for reliable evaluation. Analogously, the rise of foundation models calls for a highly diverse number of tasks. An open challenge is to decide which tasks to include in a benchmark. Adding a task to an existing set of tasks may not necessarily provide additional insights, as performance across tasks can be highly correlated (Fu et al., 2024b). From a resource investment perspective, picking the right tasks is thus highly desirable.

3. *Balancing quantity and quality is non-trivial.* Reliable reference data for robust evaluation has always been a priority. Generally speaking, increasing the number of annotators increases the quality of the reference annotation (e.g. Guzene et al. (2023), and Dumitrache et al. (2015)). However, balancing the quantity of images with the number of annotators per image is challenging. In the field of image classification, for example, prior work suggests that a single human rater is not sufficient to create reliable benchmarks (Schmarje et al., 2024). On the other hand, recent work suggests that resources may be better invested when annotating more images rather than increasing the number of annotators per item (Dorner & Hardt, 2024)]. In the context of VLM benchmarks, this issue has not yet been well-explored. According to an analysis of 13 popular VLM benchmarks, information on the number of annotators per image in VLM benchmarks is either not provided (38%), or only 3 or fewer raters (15%) contributed to creating the reference 4. Of note, MTurk has faced issues with workers producing inconsistent or careless annotations, leading to a noticeable decline in annotation quality in recent years (Kennedy et al., 2020; Rädsch et al., 2023; 2024). In general, there is insufficient evidence to determine the optimal number of annotators when considering the trade-off between annotation quality and cost.

Overall, our literature analysis clearly indicates that there is a lack of guidance on how to set up strong VLM benchmarks while keeping annotation costs low. We address this gap in the literature with the following three key contributions (see Fig. 1):

1. **Automatic task augmentation for resource-efficient creation of VLM benchmarks**: In analogy to the concept of data augmentation in traditional ML, we propose the concept of task augmentation—a resource-efficient method for generating multiple tasks from a single existing task using metadata annotations (see Fig. 2). Starting from a single task with fine-grained annotations (here: instance segmentations), metadata on each image is acquired to convert the single task to a collection of tasks (Tab. 1.), enabling comprehensible in-domain validation. To enable the augmentation, we enrich the existing datasets with metadata related to the image (e.g., number of objects, object relations, brightness) as well as individual objects (e.g., object depth, occlusion).

---

[1] lmarena.ai/?leaderboard; see the Arena(Vision) tab

[2] huggingface.co/spaces/WildVision/vision-arena

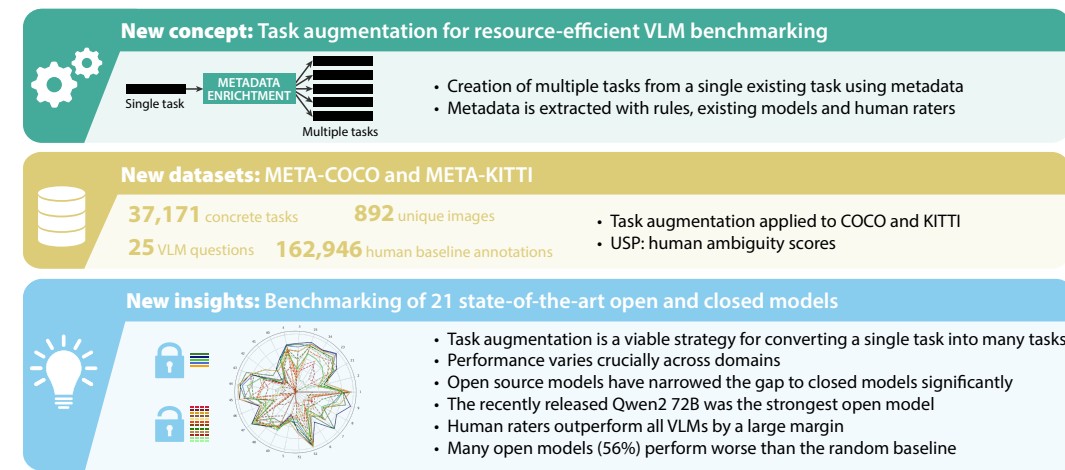

Figure 1: **Summary of contributions**. (1) New concept: We propose the concept of task augmentation as a resource-efficient method for creating multiple tasks from a single existing task using metadata annotations from multiple sources (humans, rules, models). (2) New dataset: We apply our task augmentation framework to the well-known instance segmentation tasks of COCO and KITTI to generate domain-specific VLM benchmarks with highly reliable reference data. As a unique feature compared to existing benchmarks, we quantify the ambiguity of each question for each image by acquiring human answers from a total of six raters. (3) New insights: We apply our framework to a total of 21 open and frontier closed models to demonstrate the benefit of task augmentation and to shed light on current VLM capabilities.

2. **New public data sets META-COCO and META-KITTI:** We apply our task augmentation framework to (subsets of) the well-known data sets COCO (Lin et al., 2014) and KITTI (Geiger et al., 2012) datasets to generate domain-specific VLM benchmarks with highly reliable reference data. As a unique feature compared to existing benchmarks, we quantify the ambiguity of each question for each image by acquiring human answers from a total of six raters per question. In total, this yielded 162,946 corresponding human baseline answers corresponding to 37,171 questions on 1,704 images.

3. **Comprehensive benchmarking of state-of-the-art open and closed VLMs:** We apply our framework to a total of 21 open and frontier closed models (see App. E) to demonstrate the benefit of task augmentation. Specifically, we show that the performance of models varies substantially (1) across domains, highlighting the need for domain-specific benchmarks, and (2) across tasks, indicating that our task augmentation is able to convert a single task into a diverse set of tasks suitable for VLM benchmarking. Additionally, we provide difficulty rankings for a total of 25 VLM tasks, as well as a strength-weakness analysis of existing models based on our benchmark.

## 2 RELATED WORK

### 2.1 VISION-LANGUAGE BENCHMARKS

Recent studies have proposed a range of evaluation benchmarks for VLMs, varying in size and the number and type of vision-language (VL) capabilities they assess (e.g. Fu et al. (2024b) and Bai et al. (2023)). For instance, Blink (Fu et al., 2024b) and MMBench (Liu et al., 2024a) comprise more than 3,000 multiple-choice questions and cover multiple VL tasks while MME (Fu et al., 2024a) focuses on evaluating perception and cognition abilities using Yes/No questions.

Among the largest benchmarks developed to date are MMT-Bench (Ying et al., 2024) which includes more than 31,000 questions, MME-RealWorld (Zhang et al., 2024b) comprising more than 29,000 image-question pairs, and MMMU (Yue et al., 2024) featuring 11,500 questions. While these large benchmarks cover multiple VL capabilities and domains, they require extensive labeling

efforts. For example, the authors of the MME-RealWorld benchmark involved 25 professional annotators and seven experts in VLMs, while the MMMU benchmark included contributions from 50 college students. The authors of MMT-Bench, however, do not provide the exact number of human annotators.

At the same time, MMStar (Chen et al., 2024b) and MM-Vet-v2 (Yu et al., 2024) allow the evaluation of core VL capabilities of VLMs using much smaller question sets. Additionally, several works integrate multiple existing benchmarks into comprehensive evaluation frameworks (Jiang et al., 2024; Al-Tahan et al., 2024). Platforms such as WildVision (Lu et al., 2024) and LVLM-eHub (Xu et al., 2023) enable the collection of human preferences to further enhance VLM evaluation. Other studies focus on specific aspects of VLM assessment such as accounting for uncertainty (Kostumov et al., 2024), disentangling perception and reasoning (Qiao et al., 2024), evaluating complex vision tasks (Liu et al., 2024b; Rahmanzadehgervi et al., 2024), or leveraging large language models (LLMs) to assess visual storytelling abilities of VLMs (Bai et al., 2023). Very recently, (Tong et al., 2024) presented a critical examination of multimodal LLM benchmarks.

Despite the variety of datasets and tasks, a resource-efficient and generalizable approach that enables extensive evaluation of VLMs across multiple (domain-specific) tasks is still lacking. One potential solution is task augmentation.

## 2.2 TASK AUGMENTATION AND METADATA

The challenge of generating multiple tasks from a single dataset for VLM benchmarking remains relatively unexplored. A few examples include CLEVR (Johnson et al., 2017), a benchmark dataset of 3D objects used to assess various visual reasoning tasks by combining existing metadata attributes such as the size, color, or shape of objects. "Task Me Anything" (Zhang et al., 2024a) generated an important contribution in showing the potential of automatic task generation, but focuses on computationally rendered images specifically created for the evaluation. Taskonomy (Zamir et al., 2018) proposed a fully computational approach for modeling the structure and connections within the domain of visual tasks. Other related works are LVIS-INSTRUCT4V (Wang et al., 2023), a visual instruction dataset generated with GPT-4V, and JourneyBench (Wang et al., 2024) which comprises human-annotated, generated images and requires in-depth multimodal reasoning. In addition, GoogleAI's Open Images Dataset (Kuznetsova et al., 2020) and Visual Genome (Krishna et al., 2017) can serve as a valuable resource for VLMs benchmarking as they include comprehensive pre-generated metadata. However, these datasets do not support automatic generation of metadata for other datasets, limiting their use to the specific data they contain. To the best of our knowledge, our work is the first to develop an approach for automated task augmentation from real world instance segmentation datasets using metadata enrichment.

## 2.3 RESOURCE-EFFICIENT VLM BENCHMARKING

Most existing benchmarks often focus on performance metrics without considering the computational resources required (see e.g. (Fu et al., 2024b; Liu et al., 2024a)). The work that has been done on efficient benchmarking has been only in the realm of unimodal language models (Polo et al., 2024; Perlitz et al., 2023). This is despite the fact that VLMs are becoming more prominent both in research and industry (Li et al., 2024; Yang et al., 2023) and multimodality has been described as a key measure of intelligence (Bubeck et al., 2023).

## 3 METHODS

In this section, we present the methodology related to our three core contributions (see 1).

## 3.1 TASK AUGMENTATION

The principle of task augmentation for resource-efficient in-domain benchmarking is depicted in Fig. 2. Starting from a single task with fine-grained annotations, metadata on each image is acquired from multiple sources (humans, rules, and models) to transform the single task into a collection of tasks. In this work, we use instance segmentation as the core perceptual task to generate the diverse

Figure 2: **Task augmentation using metadata.** Starting from a single task with fine-grained annotations (here: instance segmentations), metadata on each image is acquired from multiple sources (humans, rules, and models) to convert the single task into a collection of tasks. This allows for resource-efficient in-domain benchmarking of VLMs.

set of VLM benchmark tasks depicted in App. I (examples in App. A). The metadata features result from three sources:

1. Human annotators were used to generate information that cannot be extracted from the existing annotations or using established models. To this end, we outsourced annotations to a professional annotation company as well as in-house annotators. Human raters were asked to decide on whether occlusion and truncation were present in the images.

2. A rule-based approach was used to convert existing information to metadata. For example, instance segmentations were leveraged to generate the number of objects of a specific class or whether a segmentation mask was touching any other segmentation masks and, if so, which ones.

3. An existing depth foundation model, Depth Anything v2 (Yang et al., 2024), was used to generate depth maps of images.

## 3.2 META-KITTI AND META-COCO

We applied the principle of task augmentation to the well-known datasets KITTI (Geiger et al., 2012) and COCO (Lin et al., 2014). For COCO, we used the high-quality instance segmentation masks provided by COCONut (Deng et al., 2024). Overall, we added 300,000 metadata annotations to a total of 1,704 images across seven domains. This includes 15 annotations per object (e.g. occlusion, relative_size, segmask_touches_segmask, or average_depth). For truncation, occlusion, and direction, we obtained up to five annotations per object from crowdworkers (UI example is displayed in App. B). The complete list is provided in App. 5.

The metadata were then used to define a set of 25 different VLM tasks, where six concern the whole image, 13 are related to individual objects, and six concern pairs of objects.

To create a concrete list of vision-language tasks for each image we employed a systematic process. We began by prioritizing images in the datasets that featured a higher number of classes and objects to maximize task diversity and complexity. Next, specific criteria for each task were evaluated to ensure appropriate task generation for each image. For instance, for tasks that involved comparing two objects, it was essential that both objects were present in the image and belonged to the relevant classes. Furthermore, we established minimum thresholds for various measures, such as requiring a significant depth difference between objects, to ensure the correct answers for the task could be

Table 1: Summary of META-COCO and META KITTI.

| Domain | Icon | #Images | #Objects | #Tasks | Human Annotations |
|---|---|---|---|---|---|
| **Wildlife** | 🦁 | 268 | 853 | 5,528 | 24,024 |
| **Persons** | 🧍 | 250 | 7,812 | 6,122 | 26,548 |
| **Vehicles** | 🚗 | 235 | 2,199 | 5,219 | 22,976 |
| **Animals** | 🐕 | 273 | 1,162 | 5,724 | 24,907 |
| **Kitchen** | 🍽️ | 272 | 2,143 | 5,332 | 23,793 |
| **Food** | 🥦 | 236 | 5,673 | 5,249 | 23,221 |
| **Kitti** | 🐱 | 170 | 1,458 | 3,997 | 17,477 |

reliably determined. Overall, our objective was to generate as many of the 25 different tasks as possible for each image.

To rate the difficulty/ambiguity for each of the 33,174 tasks, we further acquired annotations from six human raters per image. We implemented early stopping if four raters agreed for a task. Overall, this resulted in 145,489 human reference annotations. An overview of the resulting datasets META-COCO and META-KITTI is provided in Tab. 1.

### 3.3 BENCHMARKING STRATEGY

VLM benchmarking results can vary substantially with various factors, such as the images used, the domain, and the prompts applied. This often renders comparison of results across papers infeasible. For example, Accuracy is known to be a prevalence-dependent metric, meaning that results should not be compared across datasets. To address this bottleneck, we fully homogenized our benchmarking pipeline using the proposed task augmentation concept.

Model Selection: We selected 21 frontier and open VLMs of various sizes and from various providers and sources, as illustrated in App. E. The oldest model was released in January 2024, while the most recent one was released a few days before the manuscript submission.

Benchmarking workflow: To ensure fair and consistent evaluation of all selected VLMs, we developed a standardized benchmarking workflow applied uniformly across all models. We assessed them in a zero-shot setting without any additional fine-tuning or domain-specific training. We strictly followed the configurations and setups recommended by each model's authors, using the exact settings provided in their official repositories (e.g., on Hugging Face) to ensure that each model was evaluated under conditions intended by its creators. For tasks requiring multiple images, they were combined into one. Each model was provided with a carefully crafted text prompt alongside the corresponding image(s). To eliminate potential ambiguities, we conducted iterative testing of these prompts among human evaluators. Through multiple rounds of refinement, we adjusted the prompts until all human testers consistently agreed on their interpretation.

VLM Tasks: We evaluated the models on a comprehensive set of 25 tasks derived from our task augmentation framework (examples in App. A; details in App. I). Each task was associated with specific evaluation criteria and standardized prompts. For instance, when dealing with multiple-choice questions or tasks involving object selection, we established clear guidelines on how options were presented and how objects were chosen within images. This attention to detail ensured that the evaluation was both rigorous and reproducible.

Metrics and Rankings: Choosing an adequate strategy for performance assessment is far from trivial and a research topic of its own (Maier-Hein et al., 2024; Reinke et al., 2024). In this work, we were specifically interested in relative performance differences rather than in the specific ability of models to serve a specific task. To obtain aggregated performance values across images, we introduce the Accuracy% metric. This metric is configured with a percentage threshold t. Simply put, the Accuracy%(t) metric outputs the percentage p of images, for which at least t% of questions were answered correctly.

- $D$ represents the dataset, where each image is denoted as $i \in D$.

- $M$ represents the models, where each model is denoted as $m \in M$.
- $Q_i$ represents the set of questions for image $i$.
- $C_{i,q,m}$ represents the correctness score for image $i$, question $q$, and model $m$, where $C_{i,q,m} \in \{0, 1\}$ (1 if correctly answered, 0 otherwise).
- $t \in [0, 1]$ is the threshold that specifies the desired percentage of correct answers (e.g., $t = 0.75$ for 75%).

The **Accuracy%(t)** at threshold $t$ for model m can be defined as:

$$\text{Accuracy}[\%]_m(t) = \frac{1}{|D|} \sum_{i \in D} \left( \frac{1}{|Q_i|} \sum_{q \in Q_i} I(C_{i,q,m} \geq t) \right) \times 100, \quad (1)$$

where:

- $|D|$: Total number of images in the dataset.
- $|Q_i|$: Number of questions for image $i$.
- $\sum_{q \in Q_i} C_{i,q,m}$: Number of correctly answered questions for image $i$ by model $m$.
- $\frac{1}{|Q_i|} \sum_{q \in Q_i} C_{i,q,m}$: Fraction of correctly answered questions for image $i$.

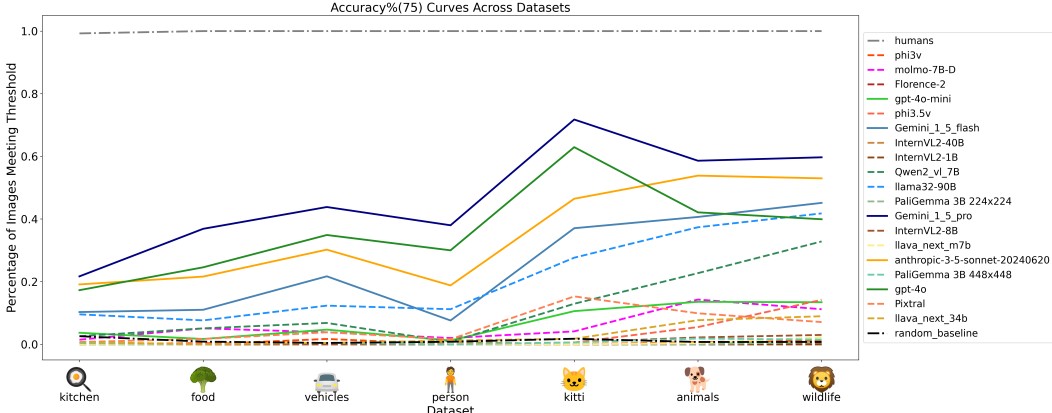

Figure 3: **The need for specific in-domain evaluation is demonstrated by the high performance variability across imaging domains.** The Accuracy%(t) metric represents the percentage of images for which at least a specified proportion of questions are correctly answered. For the best model Gemini 1.5 pro, the percentage of images for which at least 75% of (the same) questions are answered correctly varies between 22% and 72%. The full plots for all thresholds are displayed in App. D.

## 4 EXPERIMENTS AND RESULTS

The primary purpose of our experiments was to showcase the benefit of our task augmentation approach (sec. 4.1). To assess the value of each task for VLM benchmarking, we related it to average model performance, resources needed to create the task, and corresponding human ambiguity (sec. 4.2). Finally, we leveraged our concept and data to explore the capabilities of the most recent open and closed VLMs (sec. 4.3.).

### 4.1 BENEFIT OF TASK AUGMENTATION

Fig. 3 shows aggregated performance values for all models, separated by imaging domain. As the tasks and prompts were homogenized, the results clearly indicate that performance varies substantially across domains, supporting the hypothesis that in-domain validation is crucial for real-world

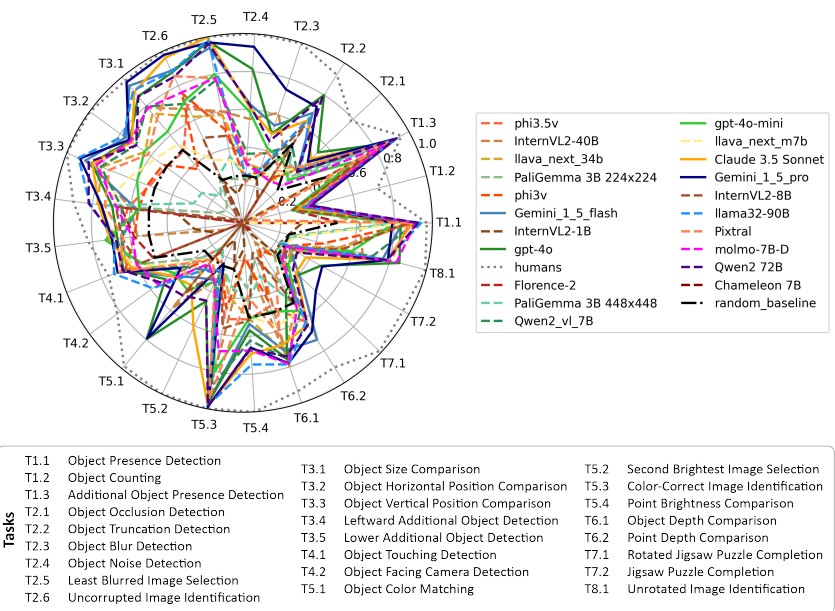

Figure 4: **Task augmentation yields a diverse set of tasks.** Spider diagram illustrating high variability across tasks. For each model and each task we aggregate the results across all datasets.

translation. Note that this holds true despite the fact that we purposely chose domains that are relatively common (presumably captured in the model training) and closely related to one another.

Furthermore, as shown in Fig. 4, the performance of models varies substantially across VLM tasks, suggesting that the tasks generated by our task augmentation approach are diverse. The hardest tasks on average across domains are (1) T7.2 "Jigsaw Puzzle Completion", (2), T1.2 "Object Counting", (3), T7.1 "Rotated Jigsaw Puzzle Completion", (4), T2.1 "Object Occlusion Detection", and (5) T5.2 "Second Brightest Image Selection". The easiest task on average was T1.3 "Additional Object Presence Detection" (see Fig. 11).

## 4.2 HUMAN AMBIGUITY

As demonstrated in App. K, there is a high discrepancy in task rankings between humans and models. While the "Jigsaw Puzzle Completion" tasks ranked amongst the most challenging for the models, humans found "Object Occlusion Detection" and "Object Touching Detection" to be the most difficult.

From a resource perspective, tasks should be (1) hard to solve for models and (2) require as little human annotation as possible. This potential trade-off is captured in App. J. It can be seen that many hard tasks, including the top four, can already be extracted from instance segmentations alone.

## 4.3 INSIGHTS ON CURRENT MODELS

Fig. 5 summarizes the performance of all models as well as the human and random baseline. The following insights can be extracted:

1. **Closed models mostly outperform open models** across tasks and domains.

2. **However, open models have narrowed the gap significantly.**

3. **Among the ones tested, Qwen2 72B is by far the best performing open model.**

4. **Large models typically outperform their smaller variants**, although exceptions exist (e.g. Molmo 7B is mostly outperforming Pixtral 12 B and Gemini Flash on occasion outperforms Gemini Pro)

5. **Human raters outperform VLMs by a large margin.** While they have close to perfect performance in the majority of tasks, they struggle with counting, occlusion and direction category tasks, with counting being the most difficult one.

## 5 DISCUSSION

This paper contributes to the advancement of VLM benchmarking in three ways.

1. Framework for domain-specific benchmarking: We showed that task augmentation, using instance segmentation as the root task, enables the generation of a diverse set of VLM tasks and could thus evolve as a core method for resource-efficient domain-specific VLM benchmarking. The insights gained on the varying difficulty of presented VLM tasks will further guide the design of future benchmarks.

2. New data: Our two new datasets META-KITTI and META-COCO will help assess generalist capabilities of future VLMs. Furthermore, we release the six human annotations per task (totaling 162,946 annotations) to assist other researchers in their benchmarking efforts.

3. New insights: The insights on current capabilities of closed and open VLMs highlight the narrowing gap between closed and open models. Most importantly, we showcased the need for domain-specific validation.

Core strengths of our contribution include the broad applicability of our concept, the open data contribution, and the wide range of state-of-the-art closed and open models investigated here, with the youngest model released only a few days before submission.

As an implicit contribution, we introduced the new metric Accuracy%, which offers several key strengths. It measures the variety of tasks on the same image, enabling a comprehensive evaluation across different capabilities, and provides a rigorous benchmark that challenges VLMs on a wide range of tasks. The metric is extendable with additional tasks, allowing for gradually increasing difficulty, and can be adapted to evaluate domain-specific tasks effectively. However, there are limitations: some questions require specific conditions, such as the presence of multiple objects for comparison, potentially resulting in variability in the number of questions per image. Additionally, tasks are treated equally without any weighting, which may overlook differences in task difficulty or importance.

A limitation of our work is model family dependence, as many models come from closely related families, which may hinder statistical analysis. For closed-source models, specific information about training and data is often unavailable, creating transparency issues. Model performance also shows prompt dependence, with results potentially varying based on prompt phrasing. Additionally, our human annotations were performed by professional annotators, which may introduce ambiguity since annotators aim to complete tasks quickly.

Future work should focus on expanding the number of tasks generated, further enhancing the diversity and comprehensiveness of VLM benchmarks. Additionally, our method can be adapted to different domains with domain-specific questions or scaled up to support continuous extension, providing a versatile approach for evaluating models across diverse applications.

### CODE

Code will be made available after acceptance.

### ACKNOWLEDGMENTS

Acknowledgments redacted for review.

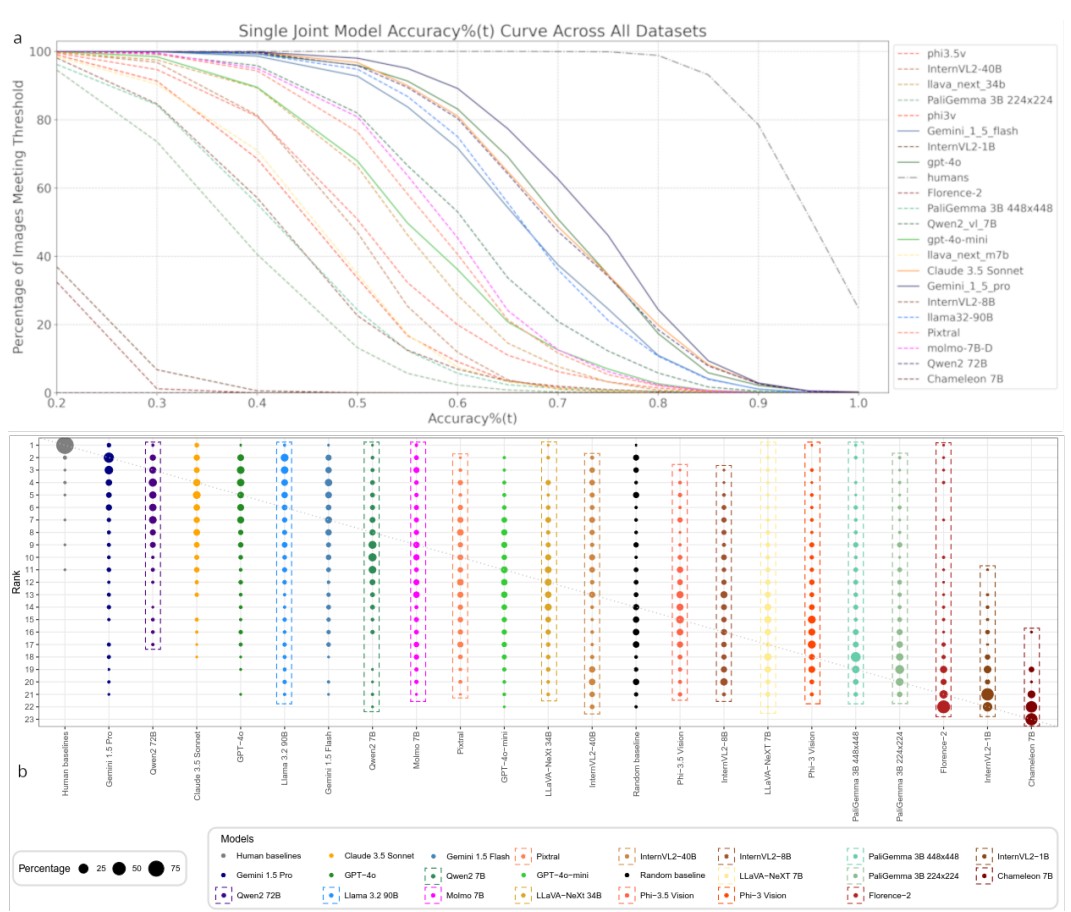

Figure 5: **Open source models have narrowed the gap to closed models significantly.** (**a**) The Accuracy%(t) metric shown across all datasets, all closed (solid line) and open (dashed lines) models investigated in this study as well as the human baseline (grey line). It represents the percentage of images for which at least a specified proportion of questions were correctly answered by the model, calculated for varying accuracy thresholds. A stratification per imaging domain is provided in App. D. (**b**) Blob plots indicating ranking variation across models over all domains. The radius of each blob at position $(M_i, rank j)$ is proportional to the relative frequency model $M_i$ achieved rank $j$ over all domains. Open models are indicated by a dashed border.

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

## A  EXAMPLE TASKS FOR AN IMAGE

Figure 6: **For each image, we generate numerous tasks covering diverse perception abilities.**

## B EXAMPLE OF HUMAN METADATA ANNOTATION

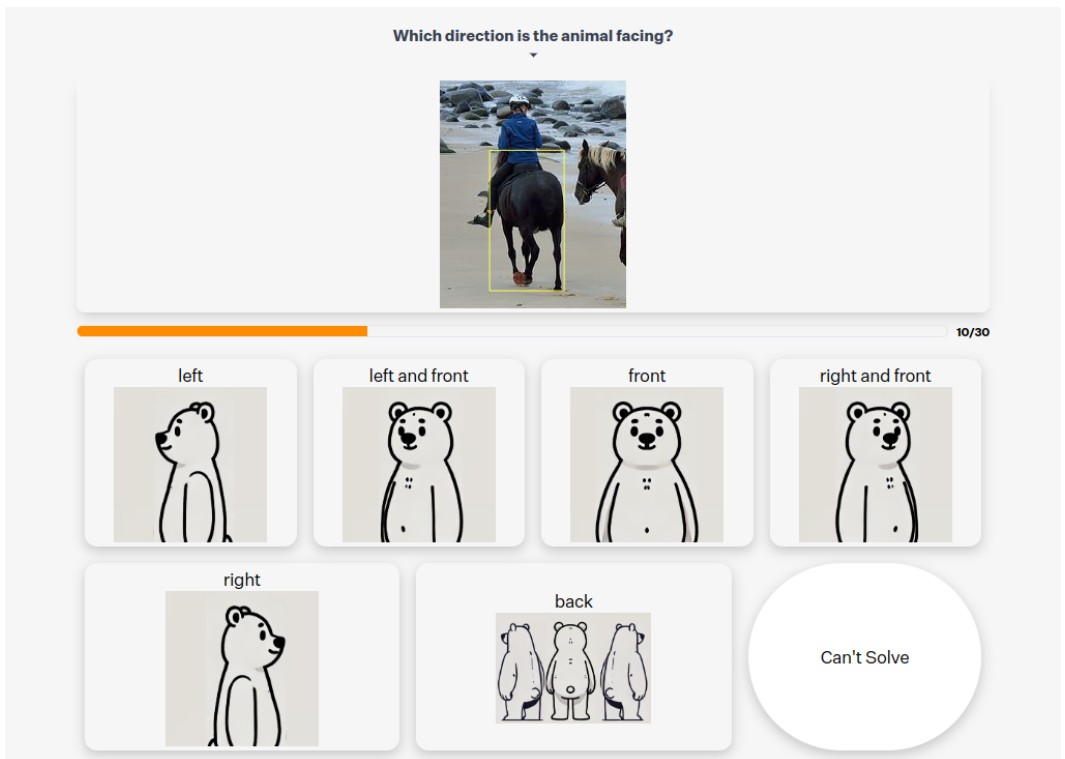

Figure 7: **Exemplary initial human metadata enrichment task.** These annotations were used to enrich the objects with human generated metadata.

## C CVPR 2024 PAPER ANALYSIS

Table 2: CVPR 2024 paper analysis summary.

| CVPR 2024 | |
|---|---|
| Total number of papers | 2,708 |
| With New or modified dataset: | 397 |
| Without new or modified dataset: | 2,311 |

We analyzed all papers from CVPR 2024 using three different large language models (LLMs). If the majority of models indicated that a paper introduced a new or modified dataset, we tagged it accordingly. This process identified 397 publications proposing a new or modified dataset. To validate the accuracy of the tagging, we randomly selected 10% of these flagged papers for a human review. All human-verified publications were confirmed to propose a new dataset.

# D   ACCURACY%(T) CURVES ACROSS DATASETS

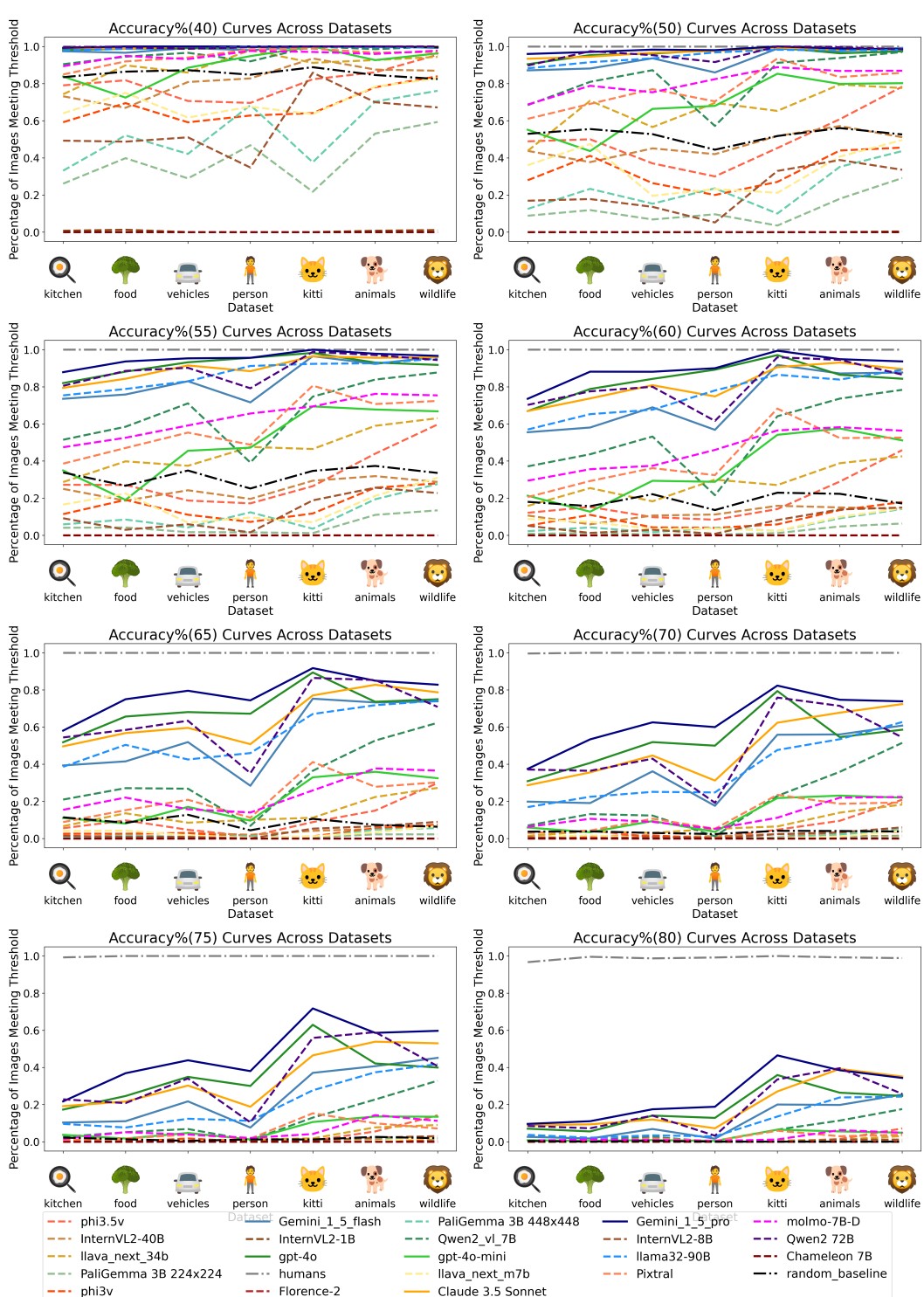

Figure 8: **The need for specific in-domain evaluation is demonstrated by the high performance variability across imaging domains.** The Accuracy%(t) metric represents the percentage of images for which at least a specified proportion of questions are correctly answered by the mode.

## E  MODEL OVERVIEW

Table 3: VLM Benchmark Models used in this study

| Accessibility | Size | Name | Version | Organization | Release Date |
|---|---|---|---|---|---|
| Closed | - | GPT-4o | gpt-4o-2024-08-06 | OpenAI | 2024-08-08 |
| Closed | - | GPT-4o-mini | gpt-4o-mini-2024-07-18 | OpenAI | 2024-07-18 |
| Closed | - | Gemini 1.5 Pro | gemini-1.5-pro-001 | Google | 2024-05-24 |
| Closed | - | Gemini 1.5 Flash | gemini-1.5-flash-001 | Google | 2024-05-24 |
| Closed | - | Claude 3.5 Sonnet | claude-3-5-sonnet-20240620 | Anthropic | 2024-06-20 |
| Open | 1B | InternVL2-1B | InternVL2-1B | OpenGVLab | 2024-07-04 |
| Open | 8B | InternVL2-8B | InternVL2-8B | OpenGVLab | 2024-07-04 |
| Open | 40B | InternVL2-40B | InternVL2-40B | OpenGVLab | 2024-07-04 |
| Open | 7B | Qwen2 7B | Qwen2-VL-7B-Instruct | Alibaba | 2024-08-30 |
| Open | 72B | Qwen2 72B | Qwen2-VL-72B-Instruct | Alibaba | 2024-08-30 |
| Open | 7B | LLaVA-NeXT 7B | llava-v1.6-mistral-7b-hf | U. of Wisconsin–Madison | 2024-01-30 |
| Open | 34B | LLaVA-NeXt 34B | lava-v1.6-34b-hf | U. of Wisconsin–Madison | 2024-01-30 |
| Open | 7B | Chameleon 7B | chameleon-7b | Meta | 2024-05-16 |
| Open | 4.2B | Phi-3 Vision | Phi-3-vision-128k-instruct | Microsoft | 2024-04-23 |
| Open | 4.2B | Phi-3.5 Vision | Phi-3.5-vision-instruct | Microsoft | 2024-08-20 |
| Open | 770M | Florence-2 | Florence-2-large-ft | Microsoft | 2024-06-15 |
| Open | 3B | PaliGemma 3B 224x224 | paligemma-3b-mix-224 | Google | 2024-05-14 |
| Open | 3B | PaliGemma 3B 448x448 | paligemma-3b-mix-448 | Google | 2024-05-14 |
| Open | 12B | Pixtral | Pixtral-12B-2409 | Mistral | 2024-09-17 |
| Open | 90B | Llama 3.2 90B | llama-3-2-90b-vision-instruct | Meta | 2024-09-25 |
| Open | 7B | Molmo 7B | Molmo-7B-D | Allen Institute for AI | 2024-09-24 |

## F  OVERVIEW OF VLM BENCHMARK ANNOTATION PROCESSES

Table 4: Overview of VLM Benchmark Annotation Processes

| Benchmark | Annotators reported? | Raters per image | Comment |
|---|---|---|---|
| Blink | Yes | 2 per image | Two annotators (co-authors) assigned per task. Exception: one question type received single annotation. |
| MMBench | No | N/A | Volunteers (students) expanded initial question set. |
| MME | No | N/A | Number of annotators unclear |
| MMStar | Yes | N/A | Three experts reviewed. Unclear if all samples seen by all. |
| MM-Vet v2 | No | N/A | GPT-4V generated drafts, experts reviewed. Exact number undisclosed. |
| MMT-Bench | Yes | 50 in total | "Dozens of co-authors" and 50 students assisted. |
| WildVision | Yes | 1 per image | Crowdsourced. Cohen's Kappa: 0.59. |
| MMMU | Yes | N/A | 50 annotators, college students from diverse disciplines. |
| II-Bench | Yes | N/A | 50 students collected and annotated images. |
| Vibe-Eval | Yes | N/A | 22 group members collected prompts. |
| TouchStone | No | N/A | Manually annotated, no statistical info provided. |
| Seed-Bench-2 | No | N/A | Partly manually annotated, number not given. |
| MME-RealWorld | Yes | N/A | 25 professional annotators, 7 MLLM experts. Task distribution unclear. |

# G   THREE METADATA SOURCES

Table 5: Metadata sources used for enriching instance segmentation datasets.

| Human Raters | |
|---|---|
| **Attribute** | **Description** |
| Occluded | Object occluded or fully visible (other object in front) |
| Truncated | Object truncated or fully visible (edge of image) |
| Direction | Direction the object is facing |
| **Existing Annotations** | |
| **Attribute** | **Description** |
| relative_size | Relative size compared to image size |
| bbox_touches_bbox | Bounding box touching another bounding box |
| segmask_touches_segmask | Segmentation mask touching another segmentation mask |
| segmask_touches_segmask_with | Specific segmentation masks touching each other |
| segmentation_area | Area covered by segmentation |
| brightness_score | Brightness score |
| michelson_contrast_score | Michelson contrast score |
| bbox_x_min, bbox_y_min, bbox_x_max, bbox_y_max | Bounding box coordinates |
| class_name | Class name of the object |
| **Model Generated** | |
| **Attribute** | **Description** |
| average_depth | Average depth of the object |
| top_95_depth | Depth of the top 95% portion of the object |
| bottom_5_depth | Depth of the bottom 5% portion of the object |

## H  MODEL ACCURACY%(T) CURVES FOR EACH DATASET

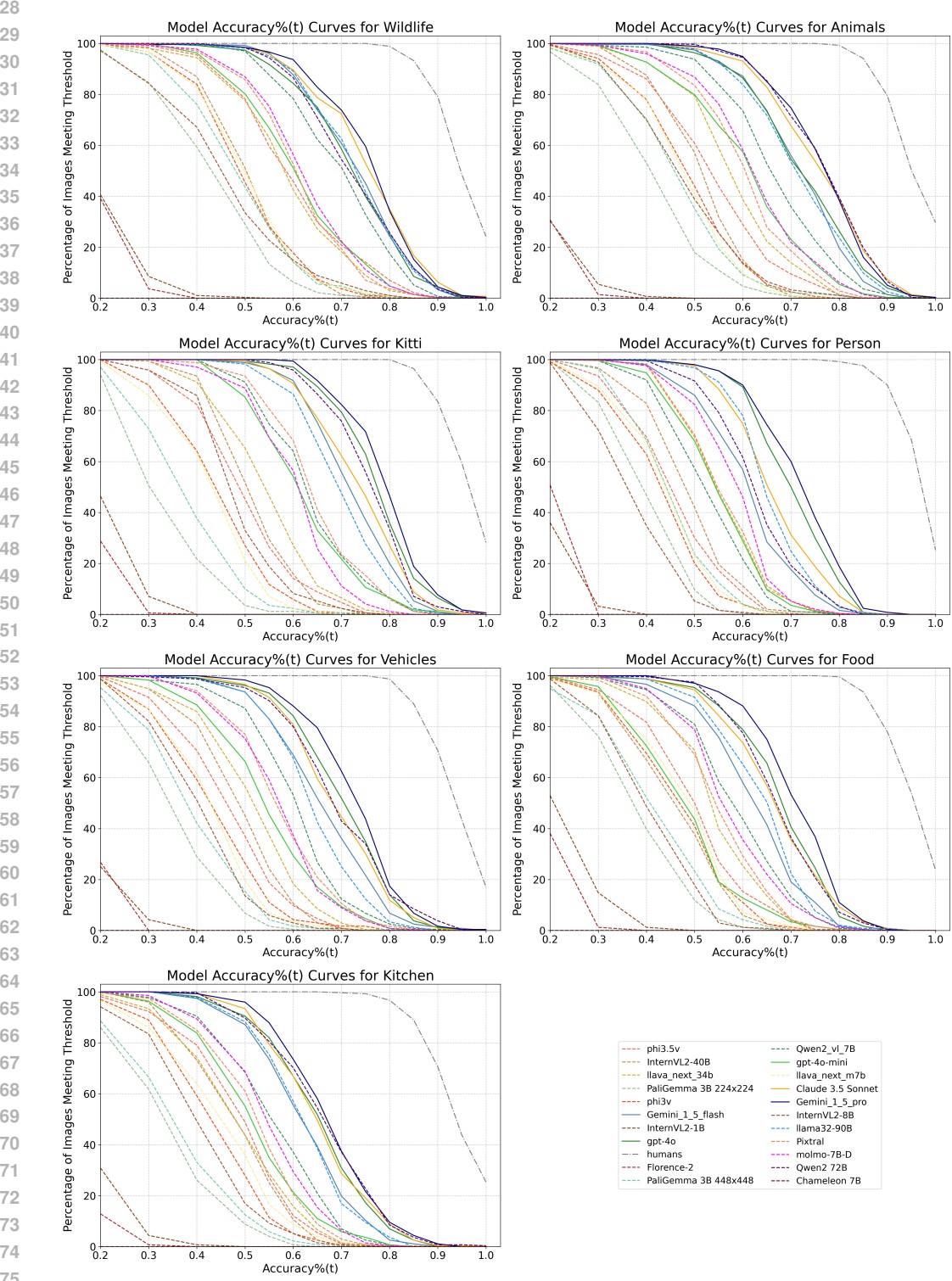

Figure 9: **Model Accuracy%(t) curves per dataset.** We observe performance variability across imaging domains and across models. Exemplary for animals, the best open model Qwen 72B (purple dashed) is almost on par with the best closed Model Gemini 1.5 pro (solid blue).

# I OVERVIEW VLM TASKS

Table 6: Overview of VLM Benchmark Tasks

| ID | Task Name | Task Description | Answer Type |
|---|---|---|---|
| T1.1 | Is Object Present | Determines whether a specified object is present in the image. | Binary |
| T1.2 | Count Objects | Determines the number of objects in the image | Count |
| T1.3 | Is Oth Object Present | Determines whether or not there is more than one object in the image | Binary |
| T2.1 | Is Object Occluded | Determines if the specified object is partially or fully occluded. | Quiz (A/B/C/D) |
| T2.2 | Is Object Truncated | Determines if the specified object is truncated in the image frame. | Binary |
| T2.3 | Blur Object | Determines whether an object is blurred | Quiz (A/B/C/D) |
| T2.4 | Noise Object | Determines whether an object contains noise | Quiz (A/B/C/D) |
| T2.5 | Blur Of Image | Determines which image variant is least blurred | Quiz (A/B/C/D) |
| T2.6 | Noise Of Image | Determines which image variant is not corrupted | Quiz (A/B/C/D) |
| T3.1 | Size Comparison | Determines which of two objects is larger | Color |
| T3.2 | Horizontal Comparison | Determines which object is further to the left of the image | Color |
| T3.3 | Vertical Comparison | Determines which object is further to the bottom of the image | Color |
| T3.4 | Is Oth Object Left | Determines whether there is another image further to the left of an object | Binary |
| T3.5 | Is Oth Object Lower | Determines whether there is another image further to the bottom of an object | Binary |
| T4.1 | Is Object Touching other Object | Determines if two objects are touching each other | Binary |
| T4.2 | Is Object Facing Camera | Determines if the object is facing the camera | Quiz (A/B/C/D) |
| T5.1 | Color Object Matching | Determines which of four tiles show the correct color for the given image | Quiz (A/B/C/D) |
| T5.2 | 2nd Brightest Image | Determines which of the images is the 2nd brightest image | Quiz (A/B/C/D) |
| T5.3 | Color Of Image | Determines which image variant is not corrupted | Quiz (A/B/C/D) |
| T5.4 | Brightness Comparison of Two Points | Determines which of two points is brighter | Binary |
| T6.1 | Depth Comparison | Determines which of two objects is closer to the camera | Color |
| T6.2 | Depth Two Points Image | Determines which point is closer | Binary |
| T7.1 | Jigsaw rotation Puzzle | Determines which of four rotated tiles fits best into a cut out area of the image | Quiz (A/B/C/D) |
| T7.2 | Jigsaw Puzzle Image | Determines which of four tiles fits best into a cut out area of the image | Quiz (A/B/C/D) |
| T8.1 | Rotation Of Image | Determines which image variant is not rotated | Quiz (A/B/C/D) |

## J  EXTRACTING HARD TASKS FROM INSTANCE SEGMENTATIONS

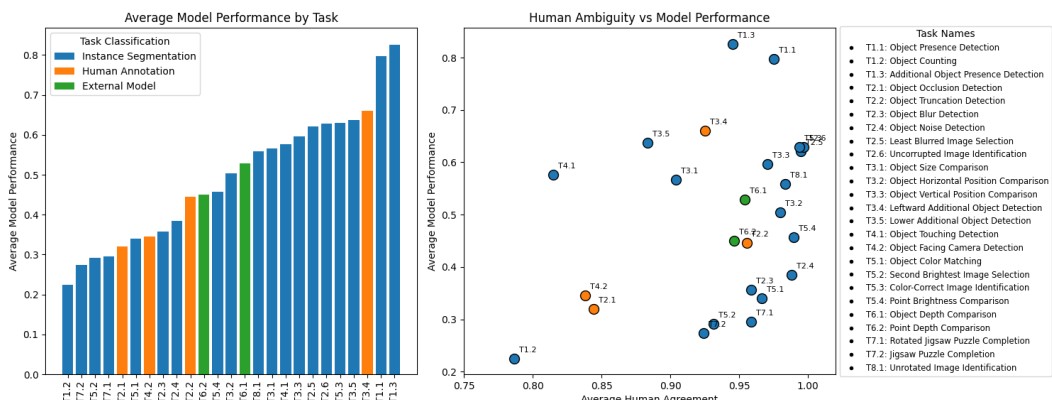

Figure 10: **Instance segmentations alone allow for the extraction of hard tasks.** (a) Tasks were classified in those extractable directly from instance segmentations (blue), requiring external models (green) and requiring human annotations (red). (b) Human ambiguity plotted against model performance.

## K  RANKING COMPARISON BETWEEN MODELS AND HUMANS

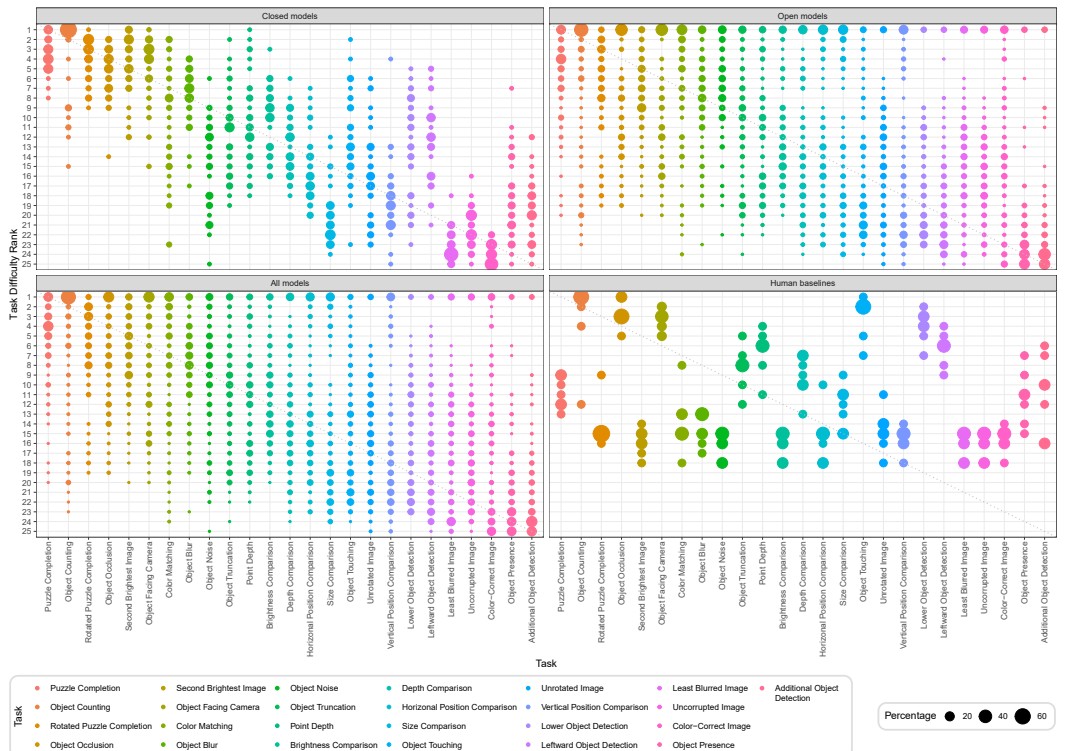

Figure 11: **Task ranking differs between models and human raters.** The plot shows the difficulty of tasks based on aggregated model scores (1 = hardest task, 25 = easiest task). The radius of the blob indicates how often a task was assigned a difficulty rank when considering all seven domains and all models (n = 5 for closed models; n = 16 for open models; n = 21 for all models; n = 1 for humans as majority vote over several raters). The larger the plot, the higher the percentage it achieved a specific rank. The hardest tasks on average across domains are (1) T7.2 "Jigsaw Puzzle Completion", (2), T1.2 "Object Counting", (3), T7.1 "Rotated Jigsaw Puzzle Completion", (4), T2.1 "Object Occlusion Detection", and (5) T5.2 "Second Brightest Image Selection". The easiest task on average was T1.3 "Additional Object Presence Detection".

