# OpenReview forum: "Domain-specific Benchmarking of Vision-Language Models: A Task Augmentation Framework Using Metadata"
_ICLR.cc/2025/Conference — ICLR 2025 Conference Withdrawn Submission_

### Official Review · Reviewer_z3t7 · 2024-10-30

**Soundness:** 3
**Presentation:** 2
**Contribution:** 2
**Rating:** 5
**Confidence:** 4

**Summary:**

This paper presents a method for repurposing  existing vision datasets to new visual tasks by leveraging the same imagery and obtaining additional metadata through a combination of human input, simple heuristic rules, and pre-trained models (e.g., segmentation and depth models). The generated data is then used to evaluate a comprehensive set of existing VLMs.

**Strengths:**

* The paper addresses a critical issue: developing evaluation datasets for domain-specific benchmarking of VLMs

* It includes an extensive evaluation using a diverse set of VLMs across various model sizes, enhancing the robustness of the findings.

* The method demonstrates effectiveness, as even powerful models struggle with some tasks, demonstrating that the generated benchmark is challenging.

* Human validation is incorporated to ensure clarity of  image-question pairs and reduce ambiguity.

**Weaknesses:**

* While the authors formalized a pipeline for “task augmentation,” the concept of repurposing existing imagery from available datasets and leveraging metadata (using off-the-shelf models or human input) to evaluate different tasks or augment training sets is well-explored in prior work. For instance, see [1],[2],[3],[4] among many others. In a way or another those benchmark repurpose existing vision datasets and use either humans or off-the-shelf models to generate additional metadata or VQA type questions.

* The paper initially frames itself as a method for generating validation data for domain-specific foundation models with predefined, specific purposes. However, most models evaluated are “generalist” VLMs rather than “specialist” models. This is fine but the motivation and message should be adjusted accordingly.  Additionally, while the motivation includes applications in fields like pathology and autonomous driving, no data or model relevant to these high-stakes areas is evaluated. Thus, the suitability of the pipeline for evaluating such specialized tasks remains uncertain.

* The writing could be further refined, as some sections take longer to convey main points. Streamlining sections such as the introduction, Section 2.2, and Section 3.3 could improve clarity and flow.

* While the proposed metric evaluation may be intuitive to the authors, incorporating more widely recognized metrics alongside individual scoring for each task could improve the benchmarks' accessibility and broader adoption.

* Some important figures, like Figure 4, are difficult to interpret due to crowding. Grouping models by parameter count or model family could help clarify these visuals. Models differing in parameter count by more than 10x may not need to be displayed together unless a significant point is being illustrated.

* In addition to releasing the code, sharing the final generated dataset could enhance its utility for the community, potentially offering greater practical value than the code alone.

Overall, I recommend that the authors improve the writing and presentation, with an emphasis on the   benchmark and findings as the main focus rather than the data generation pipeline.

[1]  Cambrian-1: A Fully Open, Vision-Centric Exploration of Multimodal LLMs
[2]  SpatialRGPT: Grounded Spatial Reasoning in Vision Language Models
[3]  Reasoning Paths with Reference Objects Elicit Quantitative Spatial Reasoning in Large Vision-Language Models
[4]  Omni3D: A Large Benchmark and Model for 3D Object Detection in the Wild

**Questions:**

see above

---

### Official Review · Reviewer_p8BK · 2024-11-03

**Soundness:** 2
**Presentation:** 3
**Contribution:** 2
**Rating:** 5
**Confidence:** 3

**Summary:**

This paper presents a domain-specific benchmark for evaluating Vision-Language Models (VLMs), utilizing a task augmentation technique. The benchmark provides interesting conclusions, such as considerable model performance variations across related domains. However, the primary contribution—the automatic and efficient task augmentation technique—warrants further examination. And some important details concerning the benchmark lack clarity. In summary, I think this work makes a valuable contribution but requires further revisions for publication.

**Strengths:**

The paper presents a new benchmark for evaluating VLMs, which contributes for the development of this field.

**Weaknesses:**

1. The core contribution, "Automatic task augmentation", as claimed in line 98, appears not to be "automatic" nor generally available. The dataset creation still involves considerable human efforts, including metadata annotation, rule-writing, task template design, and multi-round refinement of prompts (lines 308-309).
2. The concept of "Task Augmentation", although presented as new, has been thoroughly studied in previous works [1,2,3]. These works have explored methods of generating additional tasks using metadata or simple tasks for either model evaluation or instruction tuning.

**Questions:**

1. Could you offer a detailed demonstration of the human effort required in each dataset creation stage? This would help in understanding the resource-efficiency and automation of the "Automatic task augmentation" technique.
2. How does this benchmark compare to existing VLM benchmarks in terms of task quantity, question diversity, and problem difficulty? A thorough comparison would highlight the benefits of the proposed task augmentation method.
3. Can you clarify the task generation method using metadata? Is this done through pre-set question templates, generated by LLMs, or manual writing? A clear description of this would be valuable for reproduction.
4. Could you include the statistical data about the 25 tasks, such as the number of questions in each task?

[1] Luo Z, Xu C, Zhao P, et al. Wizardcoder: Empowering code large language models with evol-instruct[J]. arXiv preprint arXiv:2306.08568, 2023.
[2] Muennighoff N, Liu Q, Zebaze A, et al. Octopack: Instruction tuning code large language models[J]. arXiv preprint arXiv:2308.07124, 2023.
[3] Shypula A, Madaan A, Zeng Y, et al. Learning performance-improving code edits[J]. arXiv preprint arXiv:2302.07867, 2023.

---

### Official Review · Reviewer_KsrF · 2024-11-06

**Soundness:** 2
**Presentation:** 2
**Contribution:** 1
**Rating:** 3
**Confidence:** 5

**Summary:**

This paper proposes a new paradigm for Vision-Language Models (VLMs) by creating multiple tasks from a single existing task, called task augmentation. This is achieved by re-annotating an existing benchmark with various tools for diverse purposes. The new paradigm is validated on the COCO and KITTI datasets. Extensive experiments on the created benchmarks are conducted, giving several interesting observations.

**Strengths:**

1) The paper is well-structured and easy to follow.

2) The paper thoughtfully considers building strong domain-specific VLM benchmarks while sparing human annotation costs. I agree that picking the right tasks is challenging.

3) Building benchmarks on existing ones with re-annotations is a smart and efficient way to control data quality and diversity. The data curation pipeline may be helpful to the community.

4) Extensive evaluation results are provided. Some observations are somewhat interesting.

**Weaknesses:**

1) Although the idea is smart, the applicability of the data re-annotation pipeline is unknown. Currently,  it is demonstrated on COCO and KITTI where instance-level annotations are provided. It would be good to elaborate more about how to generalize the data generation pipeline.

2) I do not make it clear how the proposed approach can address the challenges listed in Sec.1: domain-specific validation, picking the right tasks, balancing quantity and quality.

3) The notes drawn from the evaluation results seem not new for authors. Similar conclusions can be seen in various VLM evaluation papers.

4) I do not see a reason why the proposed approach can be more useful than existing evaluation benchmarks. A detailed comparison with existing ones should be presented.

5) The paper lacks an analysis of the evaluation results or evaluation approach.

**Questions:**

$I(C_{i,q,m})$ in Eqn.(1) is not explained.

---

### Note · Authors · 2024-11-15

**Comment:**

Based on the reviewers' feedback, we are withdrawing this paper from ICLR in its current form. We thank the reviewers for their feedback and highlighting that our approach is a "smart idea" [KsrF] and our work "addresses a critical issue"[z3t7].
We will incorporate the feedback.
For clarification, the datasets, the benchmark and the annotations will be made available.
Additionally our approach is mostly automatic and the user can even remove tasks that would require additional annotations. The approach works even without instance segmentation, given that the user utilizes a prompt method, such as SAM. However that would reduce the automation level, but is still very effective.
We will ensure this information is communicated more clearly in the manuscript.
Once more, thank you for your reviewing und providing valuable feedback.

**Withdrawal Confirmation:**

I have read and agree with the venue's withdrawal policy on behalf of myself and my co-authors.